# Patients Discharged with Home Enteral Nutrition from a Third-Level Hospital in 2018

**DOI:** 10.3390/nu11112570

**Published:** 2019-10-24

**Authors:** Cristina Campos-Martín, María Dolores García-Torres, Cristina Castillo-Martín, Rocío Domínguez-Rabadán, Juana María Rabat-Restrepo

**Affiliations:** 1Endocrinology and Nutrition Department, Universitary Hospital Virgen Macarena, 41009 Seville, Spain; m_doloresgt@hotmail.com (M.D.G.-T.); rociodr91@hotmail.com (R.D.-R.); rabat@us.es (J.M.R.-R.); 2Hospital Pharmacy Department, Universitary Hospital Virgen Macarena, 41009 Seville, Spain; christinacm93@hotmail.com

**Keywords:** home enteral nutrition, nutritional support, dysphagia, stroke, head and neck tumor

## Abstract

Patients who, during admission, begin to use enteral nutrition (EN) and do not recover adequate oral intake need proper planning prior to discharge. The present study is a descriptive analysis of patients discharged with EN from our hospital in 2018. In all, the study included 141 patients (50.3% male) with an average age of 76.18 ± 14 years with the most frequent reasons for enteral support being neurological disease (71.3%) and ear, nose, and throat (ENT) and maxillofacial surgery (17.02%) (others accounted for 11.68%). In these two groups, differences were observed in both the average age (77 vs. 70.5 years) and sex of patients—mostly women (58%) in the first group and men (70%) in the second. Overall, the access routes used were nasogastric tube (76.4%), and percutaneous endoscopic gastrostomy (18.4%); 67.1% of the episodes ended by 30 June, 60.6% of patients died (47% of neurological patients), and 39.3% patients recovered function of the oral passage (85% of surgical/head and neck tumor). The duration of support was as follows: 1–3 months, 32%; 6–12 months, 26.9%; more than 12 months, 18.5%. This indicated some frequent and clearly differentiated profiles in the patients studied, which may contribute to better care and support in order to maintain long-term treatment.

## 1. Introduction

Disease-related malnutrition is an important aspect to consider in the integral care of hospitalized patients, due to its prevalence and consequences [1]. Whether the admitted patient presents with malnutrition at the beginning of the hospital stay as a result of a chronic or acute process, or whether malnutrition develops during admission due to the course of the pathological process or treatments applied, a nutritional assessment should be carried out, and artificial nutritional support should be installed when necessary [2]. Depending on the situation, this support may be withdrawn during admission or continued after hospital discharge at the patient’s residence.

Enteral nutrition is the administration of chemically defined formulas by tube to the functioning digestive tract when oral feeding is not possible or sufficient. Continuing this support after discharge allows patients to go home once a hospital stay is not necessary for other reasons, thereby reducing the length of stay and associated health costs [3].

In Spain, the cost of home enteral nutrition is fully covered by the Spanish National Health System in accordance with the defining criteria of the products eligible for public financing and the pathologies contemplated. This is regulated by Royal Decree 1030/2006, which makes it valid throughout the whole national territory [4]. Despite this advantage, patients who are able to feed orally before admission but return home requiring enteral nutrition experience great changes in their domestic routine, and hospital professionals must provide the essential tools to make this transition as bearable as possible.

The continuation of enteral nutritional support at home should be planned in advance of hospital discharge once the access route is available and the chosen management guidelines and type of feed have been established and well tolerated. In order for patients to be nutritionally self-sufficient at home, they and/or their caregivers must be instructed in the handling of enteral nutrition, including the supply of food, the consumable materials, and the care plan. After that, they can be discharged if there is no other reason to extend the hospital stay [5,6].

The aim of this work was to collect and analyze the characteristics of enteral nutrition (EN) patients admitted to different wards of our hospital, from a base population of 481,000 patients, in 2018. More specifically, we focused on EN patients who began nutritional support during admission and continued this support after discharge.

## 2. Materials and Methods 

The selected patients were adults admitted to different wards of the Virgen Macarena University Hospital between 1 January and 31 December 2018, for whom enteral nutritional support was initiated during admission and maintained after discharge. The exclusion criteria were as follows: patients who recovered oral tolerance prior to discharge; patients who were transferred to another hospital and did not go home; and patients who entered with enteral nutrition already established from primary care, medical nutrition consults, or previous hospital admissions.

We carried out a retrospective descriptive study of the clinical characteristics of this cohort of patients and the enteral nutritional support with which they were discharged during the mentioned period. The patients were followed up until 30 June 2019. All recruited subjects gave informed consent for the recording and use of their clinical data before being included in the study. The study was conducted in accordance with the Declaration of Helsinki.

We studied epidemiological variables (age and sex), nutritional assessment parameters using the Malnutrition Universal Screening Tool (MUST) scale, and aspects of the enteral nutrition received, including indication, access route, type of formula, caloric contribution, support, and duration. A statistical analysis was carried out using T-tests and descriptive measures, which were calculated with Statistical Package for the Social Sciences SPSS ^®^ (SPSS Inc, IBM, Chicago, IL, USA) version 21.0.

## 3. Results

A total of 141 inpatients admitted to the hospital without prior artificial nutritional support and discharged home with enteral nutrition were included in the study. The sex distribution of the selected patients was balanced, with 50.3% men (*n* = 71) and 49.6% women (*n* = 70). However, a wide range of ages were included, with an average age of 76.18 ± 14 years (range: 33 to 101 years old) and a significant difference (*p* = 0.013) between the average age of men (75 years old) and women (84 years old). The MUST score at the time of nutritional screening was 2 for 43.1% of the patients, 3 for 28%, and 4–6 for 28%.

Among the diagnoses causing the implementation of home enteral nutrition, the most common was neurological disease with aphagia or severe dysphagia (71.6% of the patients), followed by ear, nose, and throat (ENT) and maxillofacial surgery (17.1%), head and neck tumors (7%), digestive tumors (3.5%), and nontumor esophageal stricture (0.7%) (Figure 1). Comparing the ages of patients with an EN probe indication because of neurological disease with those due to ENT and maxillofacial surgery and those with head and neck tumors, we found that patients with neurological disease were older (77 vs. 70.5 years, *p* < 0.001) with a greater percentage of females (58%). Patients who underwent otolaryngology and maxillofacial surgery tended to be male (70%) (Table 1). 

The most frequent administration route was nasogastric tube (NGT, 76.4%), followed by percutaneous endoscopic gastrostomy (PEG, 18.4%), jejunostomy tube (2.1%), and nasojejunal tube (1.4%). Of the 110 patients discharged with NGT, five were subsequently admitted for placement of a PEG tube.

A total of 29 different EN formulas were prescribed to the patients at discharge. They were classified into four categories according to composition: standard (34.7%), including formulas with different types and proportions of fiber in their composition; pathology specific (31.3%), including enteral feeding for hyperglycemia/diabetes and oligomeric formulas for gastrointestinal dysfunction, immunonutrition, renal insufficiency, or sarcopenia/fragility; hyperproteic formulas (26.4%); and hypercaloric formulas (5.6%). The total amount of estimated calories prescribed ranged from 1000 to 2800 kcal per day, contributing volumes ranging from 800 to 2000 mL of formula per day. The majority of patients (54.2%) were prescribed 1500 mL of enteral feed every 24 h at discharge.

In all, 94 episodes of home enteral nutrition (67.1% of the total) were documented in the follow-up period between the start of nutrition at home and 30 June 2019. The rest of the patients continued with EN. The reasons for withdrawing nutritional support were as follows: 57 patients (60.6%) died, and 37 patients (39.3%) had recovery of the oral route. Among the patients with neurological disease, 56% ended enteral support, 9% of them by recovering passage to the oral route and the rest by death. Of the patients with EN due to surgery or a head and neck tumor, 94% finished enteral support; among them, 85% recovered the oral route and 8.8% died (Table 1). 

Regarding the duration of nutritional support at home, 32% of patients needed EN for 1 to 3 months, 26.9% for 6 to 12 months, 18.5% for more than 12 months, 14.1% for 3 to 6 months, and 8.5% for less than a month (period between onset of EN in 2018 and data collection on 30 June 2019).

## 4. Discussion

In this study, we broadly defined two distinct predominant profiles, based on those registered in other hospital series [7,8,9] and home enteral nutrition (HEN) programs [10], as well as in the NADYA (home and ambulatory artificial nutrition group) Spanish HEN registry, voluntary and nationwide [11]. The two types of EN patients after hospital discharge are those with neurological diseases and those with a neoplastic process, either a head and neck tumor after ENT or maxillofacial surgery, or a digestive tumor that prevents oral intake.

Stroke is a major cause of disability and malnutrition [12] and was a pathology for one-third of our patients with EN due to neurological disease. Dysphagia is one of its most frequent sequelae, resulting in decreased food intake and dehydration, thereby increasing the likelihood of morbidity and mortality [13,14,15]. A nasogastric tube is the route of choice for enteral nutrition in the acute phase of stroke, and if its withdrawal is not possible in a reasonable period of time, PEG must be considered. In the case of amyotrophic lateral sclerosis (ALS), due to its characteristics, it is recommended to evaluate PEG in the early stages when the safety and efficacy of swallowing begin to be compromised [16]. In our 2018 series, 12% of PEG tubes were placed in patients with ALS. Most of the patients included in this study with tubes due to neurogenic dysphagia were elderly and had different pathologies causing cognitive impairment. In these patients, the decision to initiate EN should be made by assessing risks and benefits, as better results are produced after careful selection of patients [17].

The other large group of patients was those with neoplastic head and neck pathologies; almost all of these pathologies occurred post-surgery or following the development of a tumor of the upper digestive system in a small group of patients. During the surgical procedure, a nasogastric tube is placed in order to start enteral feeding as early as possible, and it is maintained after hospital discharge. In our study, most of the patients were male, younger than the neurological patients, and had recovered oral intake. In spite of the recommendation for a prophylactic PEG in some guidelines [18], the nasogastric tube has a predominant role due to its low rate of complications. In the case of gastric tumor surgery, jejunostomy tubes were placed, which were maintained at discharge when the oral intake had not recovered sufficiently.

In this cohort of patients, approximately half maintained EN for more than 6 months, and almost a fifth for more than 12 months. Most were patients with neurological disease. Different studies have shown that, with long-term treatment, this type of patient would benefit from a multidisciplinary home support team focused on EN to optimize resources, reduce the number of required hospital visits, and ensure better patient care [19]. Although continuity of care and contact with the nutrition unit are assured before discharge, as well as during follow-up in coordination with primary care, this study shows some areas for improvement, which could enhance the quality of life of our patients and their caregivers.

The analysis of the data corresponding to 2018 provides us with an overview of the reality of patients who initiate enteral nutritional support at home after hospital discharge in our area at the present time. Continuing the registration successively in the coming years will provide information to help us develop a framework within which we can highlight the evolution of needs and identify aspects that could be improved in the care of our patients.

## 5. Conclusions

In this study, the most frequent diagnosis in patients discharged with HEN was neurological disease causing aphagia or severe dysphagia, followed by oncological pathologies (head and neck or digestive cancer). NGT was the preferred type of enteral tube. 

The most commonly used EN formulas were of the standard type, followed closely by pathology-specific formulas.

Patients with neurological pathologies required HEN for longer periods of time: 56.4% had EN support for more than 6 months and 22.7% for more than a year. 

The primary reason for withdrawing home nutritional support was the death of the patient.

In conclusion, we have defined the profiles of our patients receiving tube feeding at discharge and those who would benefit from improvements in home support.

## Figures and Tables

**Figure 1 nutrients-11-02570-f001:**
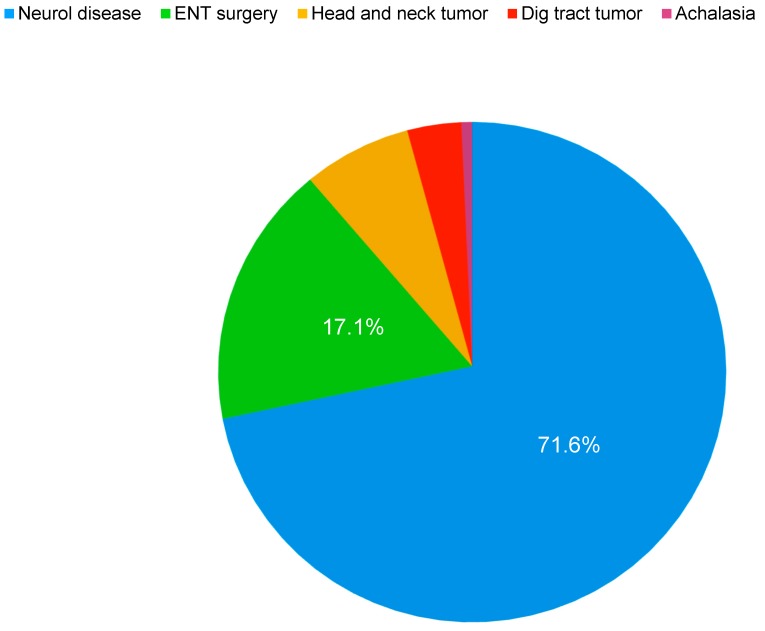
Diagnoses of patients requiring home enteral nutrition (HEN).

**Table 1 nutrients-11-02570-t001:** Characteristics by pathology. ENT, ear, nose, and throat; EN, enteral nutrition.

	Neurological Disease	ENT Surgery/Head and Neck Tumors
**Age (years)**	77	70
**Gender**	58% female	70% male
**Finished EN**	56%	94%
**Reason for withdrawal: deceased**	47%	8.8%
**Reason for withdrawal: recovered oral feeding**	9%	85%
**EN > 3 months**	72.2%	17.6%

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
