# Peer review of "Patients Discharged with Home Enteral Nutrition from a Third-Level Hospital in 2018"

_nutrients, 2019, doi:10.3390/nu11112570_

Round 1

Reviewer 1 Report

Thanks for the opportunity to review. I would suggest extensive editing be done for style and English language.  It is difficult to review as written.

Author Response

Dear Sr/Mrs,

the article has been fully edited as your request. I hope you will finally be able to review it.

Thank you very much for your time and advices. 

Regards, Cristina Campos. 

Reviewer 2 Report

This manuscript reports of Spanish patients who administered with oral intake and discharged with enteral nutrition. The issue of enteral nutrition has cons and pros, and it's getting more severe problems considering our aging society. 

However, this manuscript describes only for 140 patients in 2017, followed up only for a half year, which results in small information. They also showed statistically significant difference between the reasons of ET, gender, reason for withdrawal, and recovery of oral feeding. I am hesitant to say, however, these informations are already published from another institutions, and the results are very plausible considering the difference of neurological disease or ENT surgery including head and neck tumors. 

Author Response

Dear Sr/Mrs

the article has been fully edited as your request. 

Thank you very much for your feedback. 

We have recorded patients with HEN at discharge for +10 years, but the aim was to study the profile of patients we are attending right now. Due to the age of the patients, and their comorbidities, patients we attended years ago may have passed away and we would not have the possibility to contact them to check any missing data. On the other hand, if we want to develop improvements we need to know the current situation. 

I enclose the enhanced version. Please see the attachment,

Thank you very much for your time and suggestions

Regards, Cristina Campos. 

Reviewer 3 Report

The manuscript deals with an interesting issue, the home enteral nutritional (EN) support in dicharged patients.

The Authors conducted a retrospective descriptive analysis of a good sample of patients who started enteral nutritional support during admission and maintained after hospital discharge.

Data were collected and exposed clearly. The discussion is well articulated according to the results. The Authors broadly defined two distinct profiles of patients discharged with home EN support: patients with neurological disease (the major part) and patients with a neoplastic process (either a head and neck tumour, after ear/nose/throat and maxillofacial surgery, or a digestive tumour that prevents oral intake) and data regarding these two cohorts were scrutinized.

The English is basic and simple but correct and the table / figures are representative. Limitations of the study are described.

Although the study does not afford to derive any guideline about the management and the choice of the type of home EN support, it provides a good overview about this issue.

Author Response

Dear Sr/Mrs,

the article has been fully edited of English language and style. 

The aim of our study was to know the profile of patients at discharge with HEN and indentify those who would benefit more from improvements in home care.

Thank you very much for your time and feedback

Please see the attachment file.

Regards, Cristina Campos. 

Round 2

Reviewer 1 Report

Thanks for revising the paper. It is much easier to follow.

a couple of suggestions for the conclusion:

Change the most used EN formula to The most commonly used EN formulas Change the main reason for withdrawing to The primary reason for withdrawing.

Author Response

Dear Sr/Mrs:

I do truly appreciate your suggestions and comments. I include the latest version with the proposed changes.

Thank you very much for your valued time

Regards, Cristina Campos

Reviewer 2 Report

Your manuscript had been well reconsidered and modified.

Author Response

Dear Sr/Mrs:

Thank you very much for your review and comments. 

I include the latest version.

Regards, Cristina Campos. 
